# Anaplastic Thyroid Carcinoma: An Update

**DOI:** 10.3390/cancers14041061

**Published:** 2022-02-19

**Authors:** Arnaud Jannin, Alexandre Escande, Abir Al Ghuzlan, Pierre Blanchard, Dana Hartl, Benjamin Chevalier, Frédéric Deschamps, Livia Lamartina, Ludovic Lacroix, Corinne Dupuy, Eric Baudin, Christine Do Cao, Julien Hadoux

**Affiliations:** 1Department of Endocrinology, Diabetology, Metabolism and Nutrition, Lille University Hospital, 59000 Lille, France; arnaud.jannin@chu-lille.fr (A.J.); benjamin.chevalier@chu-lille.fr (B.C.); christine.docao@chru-lille.fr (C.D.C.); 2H. Warembourg School of Medicine, University of Lille, 59000 Lille, France; a-escande@o-lambret.fr; 3Academic Radiation Oncology Department, Oscar Lambret Center, 59000 Lille, France; 4Cancer Medical Pathology and Biology Department, Institute Gustave Roussy, 94805 Villejuif, France; abir.alghuzlan@gustaveroussy.fr; 5Department of Radiation Oncology, Institute Gustave Roussy, Université Paris Saclay, 94805 Villejuif, France; pierre.blanchard@gustaveroussy.fr; 6Département d’Anesthésie, Chirurgie et Interventionnel (DACI), Institute Gustave Roussy, Université Paris Saclay, 94805 Villejuif, France; dana.hartl@gustaveroussy.fr; 7Department of Head and Neck Oncology, Institute Gustave Roussy, Université Paris Saclay, 94805 Paris, France; frederic.deschamps@gustaveroussy.fr; 8Cancer Medicine Department, Institute Gustave Roussy, Université Paris Saclay, 94805 Villejuif, France; livia.lamartina@gustaveroussy.fr (L.L.); eric.baudin@gustaveroussy.fr (E.B.); 9Department of Medical Oncology, Institute Gustave Roussy, Université Paris Saclay, 94805 Villejuif, France; ludovic.lacroix@gustaveroussy.fr; 10CNRS, UMR 9019, 94805 Villejuif, France; corinne.dupuy@gustaveroussy.fr

**Keywords:** anaplastic thyroid carcinoma, chemotherapy, immune checkpoint inhibitors, tumors associated macrophages, radiotherapy, molecular targeted therapy

## Abstract

**Simple Summary:**

Anaplastic thyroid carcinoma (ATC) has a dismal prognostic. Chemotherapy and radiotherapy are the mainstem options for patients with ATC. In selected cases with actionable genomic alterations or with favorable immune tumor microenvironment, new therapeutic options as targeted therapies and immunotherapy have led to better outcome and raised some hope for treatment of this deadly disease.

**Abstract:**

Anaplastic thyroid carcinoma (ATC) is a rare and undifferentiated form of thyroid cancer. Its prognosis is poor: the median overall survival (OS) of patients varies from 4 to 10 months after diagnosis. However, a doubling of the OS time may be possible owing to a more systematic use of molecular tests for targeted therapies and integration of fast-track dedicated care pathways for these patients in tertiary centers. The diagnostic confirmation, if needed, requires an urgent biopsy reread by an expert pathologist with additional immunohistochemical and molecular analyses. Therapeutic management, defined in multidisciplinary meetings, respecting the patient’s choice, must start within days following diagnosis. For localized disease diagnosed after primary surgical treatment, adjuvant chemo-radiotherapy is recommended. In the event of locally advanced or metastatic disease, the prognosis is very poor. Treatment should then involve chemotherapy or targeted therapy and decompressive cervical radiotherapy. Here we will review current knowledge on ATC and provide perspectives to improve the management of this deadly disease.

## 1. Introduction

Anaplastic thyroid cancer (ATC) is a rare malignancy with a poor prognosis. It is characterized by a rapid onset with local and distant metastases, local progression and distant evolution [1]. Its treatment is an emergency, based on surgery (if feasible), radiation therapy and chemotherapy [2,3]. However, the prognosis remains very poor with a one-year overall survival (OS) rate between only 20 and 50% [4,5,6,7]. Recent results of the dabrafenib/trametinib combination for *BRAF*-mutated patients [8,9] and of immune checkpoint inhibitors (ICI), used alone or in combination with targeted therapies [10,11,12,13], have raised some hope toward an improvement of the prognosis of this deadly disease [14].

Here we will review current knowledge on the epidemiology, pathology/biology and standard treatment of ATC and discuss the recent progress and perspectives for their management.

## 2. Epidemiology and Clinical Presentation: A Rare Disease with a Rapid Onset and Poor Prognosis

ATC is a rare cancer as defined by the European Union rare cancer surveillance program (RARECARE) with an incidence far below six new cases per 100,000 person-years [15]. Indeed, recent epidemiological studies have confirmed an age-adjusted incidence in the US of 0.12 per 100,000 person-years (95% CI: 0.8–1.6) in 2014 in the Surveillance, Epidemiology, and End Results (SEER) database [16] and 0.1 to 0.3 per 100,000 person-years in Europe according to data from Denmark [4], Wales [17] and the Netherlands [18] registries. The SEER database and the Netherlands registries analyses have suggested an increase in the age-adjusted incidence with an average annual percent change of 3.0% per year (95% CI: 2.2–3.7%) and 1.3% per year (95% CI: 0.4–2.1%), respectively, over a 30–40-year period of time; this increase in incidence was not reported in the Danish and the Welsh databases [4,16,17,18]. The reasons for the discrepancies are unknown; however, incidence rates are consistent across the different studies in Europe and US. The increase in incidence is unlikely to be related to better screening/diagnosis because all ATC patients end up being diagnosed with cervical compressive symptoms.

ATC usually affects elderly patients, with the majority being over 60 years old with a female predominance (male/female sex ratio = 1.5:2). The advanced stage of the disease is the most common diagnosis presentation (localized (IVa) 10%, locally advanced (IVb) 35% and metastatic (IVc) 55%) displaying extremely aggressive behavior with rapid tumor progression, local invasion and/or distant metastases (lung, bone, liver and/or brain metastases) [1,2,19,20,21] (Figure 1).

Classically, patients report a rapid transformation of a long-standing goiter (30% of cases) within a few days to a few weeks. The others symptoms include neck pain as well as signs of neck tumoral invasion and compression: dyspnea, dysphonia and dysphagia.

On clinical examination, there is a large cervical mass, hardness on palpation, palpable lymphadenopathy and sometimes skin involvement. Invasion of the aerodigestive tract is frequent. Laryngeal dyspnea, dysphagia or superior vena cava syndrome which may require urgent treatment (placement of a tracheotomy and/or placement of a gastrostomy) as well as local pain can be associated. The repercussions of local tumor spread on the general condition immediately indicate the gravity of the situation.

## 3. Pathology and Biology: How Do We Understand the Aggressiveness of This Disease?

### 3.1. Pathology

ATC is defined as a highly malignant tumor composed of undifferentiated cells which retain some features of an epithelial origin on morphology and/or immunohistochemical examination [22]. Various and heterogeneous histologic features can be seen in ATC samples including epithelioid and squamous morphology, giant cells, pleomorphic morphology, osteoclast giant cell-rich morphology, and spindle cell morphology which is the most common histotype [23]. The most recent series from the Memorial Sloan Kettering Cancer Center (MKKCC) has examined the clinic–pathologic features of 360 cases from two institutions over a 34-year period. In this study, the most common histological subtypes were spindle cell (26%), pleomorphic (23%) and squamous (21% of the cases) [7]. Tumor necrosis was found in 77% of the cases, atypical mitosis in 77% and a neutrophilic infiltrate was noted in 71% of the cases. Interestingly, the mitotic index was >20 mitoses per 10 high-power fields in only 15% of the cases, and Ki67 was not reported. This study confirmed that thyroglobulin and TTF1 immunohistochemistry is almost always negative (96 and 70% of the cases, respectively), whereas cytokeratins AE1/AE3 are present in 67% of the cases and PAX 8 in up to 70% (with anti-PAX8 antibody 10336-1-AP). A recent immunohistochemical study, with the most commonly used monoclonal anti-PAX8 antibody (MRQ-50), showed lower PAX8 expression in 54.4% of the ATC cases [24]. Therefore, performing PAX8 immunohistochemistry in all samples of thyroid undifferentiated tumors suspicious for ATC, and in particular in squamous subtypes, allows for support of a differential diagnosis with squamous cell carcinoma of the head and neck which is always negative for PAX8 [25].

ATC tumorigenesis may be a multistep process with a biological transformation (synchronous or metachronous) from differentiated thyroid cancer (DTC) to ATC. This assumption is suggested by the common recognition of a concomitant DTC tumor component or a history of DTC observed in 58 to up to 90% of cases [7,26]. Poorly differentiated DTC (PDTC) and the tall cell variant of papillary DTC were the most common subtypes found associated to these transformed ATC subtypes in the MSKCC series [7]. From a molecular point of view the association of additional *T**P53* and/or *TERT* promoter mutations is found in up to 80% of ATC cases harboring typical DTC molecular alteration in the *BRAF* and *RAS* genes [27,28,29,30]. These data have suggested that additional mutations in *TP53* and *TERT* may drive the tumor progression from DTC to ATC [1] in these transformed ATC subtypes. This transformation process of ATC may occur differently according to the genetic mutation background. Indeed, in *RAS* mutant ATC, a history or concomitant DTC is observed in 38% of cases and it is observed in 75% of cases with *BRAF* mutations (*p* = 0.001) [7]. A whole-exome sequencing analysis of the two tumor components (DTC and ATC) of three mixed ATC tumor samples revealed that most of the somatic mutations identified in the ATC component differed from the ones in papillary DTC. This led to the conclusion of there being very few common mutations and a large genomic divergence between the two components challenging the concept of tumor progression from DTC to ATC [31]. From the clinical point of view, a recent retrospective multicenter and SEER database study on 642 primary (i.e., tumors with no DTC component at diagnosis) and 47 secondary ATC (i.e., tumors with a DTC component at diagnosis), found no statistical differences in terms of demographic, clinical manifestations and patient survival and a more frequent *BRAF* mutation as compared with *RAS* mutation in secondary tumors [32]. However, it must be pointed out that identification of “transformed” ATC requires the knowledge of the detailed clinical history of the patient as well as the detailed pathology assessment of the tumor which may not be optimal in a very large database. Therefore, it is still not known whether primary or “pure” ATC carries a different prognosis as compared with secondary or “transformed” ATC.

### 3.2. Molecular Biology

In a recent series of 126 samples of ATC analyzed by Next Generation Sequencing (NGS), the most common molecular alterations were found in *TERT* promoter (75%), *TP 53* (63%), *BRAF* (45%), *RAS* (22%), *PIK3CA* (18%), *EIF1AX* (14%) and *PTEN* (14%) with the first two being more frequent in ATC than the others, which can be seen in either DTC or PDTC [7,28]. Strikingly, *BRAF* mutation frequency in ATC seems to differ between recent series for US and Europe. Indeed, *BRAF* mutations are found in 40–45% of cases in US studies [7,14] whereas they are found in 14–37% of cases in European studies [27,30,33]; furthermore, data from south Korea reported a rate of 41% *BRAF* alterations in a series of 13 ATC cases [34]. Whether these discrepancies are linked to various sequencing techniques and/or to geographical differences in pathophysiology remains an open question. Recent results have revealed that *NTRK* and *RET* fusion can be detected in 2–3% of ATC cases [7,35,36,37] which is of utmost importance for the few patients who may be offered highly specific targeted therapies. If regulation of cell cycle has a crucial role in oncogenesis and particularly in ATC, protein metabolism control is also involved in tumorigenesis. For example, about 10% of patients with ATC harbor *EIF1AX* mutations, which has recently been involved in deregulating protein synthesis [36]. Interestingly, *EIF1AX* mutations could co-occur with *RAS* mutations in ATC with a positive feedback relationship between RAS and EIF1AX proteins, which reinforces *c-MYC* gene expression [28,38]. Molecular alteration of the Wnt signaling pathway could also be observed, notably with β-catenin gene (*CTNNB1*), *AXIN1* and *APC mutations* [36]. Although increased levels of cytoplasmic β-catenin are observed in most thyroid cancer cells, mutations of β-catenin that lead to nuclear localization of the protein are limited to PDTC and ATC, suggesting a role in tumor progression [39]. Alterations of epigenetic-related genes such as the chromatin remodeling SWI/SNF complex (*ARID1A*, *SMARCB1*, *PBRM1*, etc.) and histone methyltransferases (*KMT2A*, *KMT2C*, *KMT2D* and *SETD2)* have been found in 36% and 24% of ATC samples, respectively. The DNA Mismatch repair (MMR) pathway may be altered in 10–15% of cases [7,28,29,40,41,42,43].

Knowledge of the molecular alterations in ATC patients has been more and more prominently important in the clinics recently, with the advent of targeted therapies. This growing importance has been recognized in the recent ATA and ESMO guidelines which recommends offering molecular testing to all ATC patients with unresectable disease [2,3].

### 3.3. Immune Infiltrate of ATC

ATC tumors are characterized by an important infiltration of Tumor-Associated Macrophages (TAMs) which can represent 40 to 70% of the total tumor mass and could play some role as an immunosuppressive tumor stroma, in treatment resistance and in the poor prognosis of the disease [44,45,46,47]. ATCs display a very dense network of interconnected “ramified” TAMs which may have metabolic and trophic functions via direct contact with intermingled cancer cells [44]. This macrophage infiltration is composed of M2 pro-tumorigenic tumor-associated macrophages as demonstrated by the identification of the M2-TAMs transcriptomic signature of 78 genes identified in ATC samples, which is able to discriminate them from DTC samples [28]. The co-culture of thyroid cancer cell lines with M2-like TAMs facilitates dedifferentiation, proliferation, migration and invasion in thyroid cancer cells through the Wnt/ß-catenin pathway activation by Wnt1 and Wnt3a secretion [48] but also through insulin-like growth factor (IGF) secretion which promotes thyroid cancer stemness and metastasis by activating the PI3K/AKT/mTOR pathway [48]. A study of 19 ATC samples evaluated by automated digital quantification of CD68 and CD163 immunohistochemistry positivity confirmed the putative importance of macrophage infiltration. The mean macrophage infiltration rate was 17% and 23% for these two markers, respectively, and most of the ATC samples displayed a low to moderate level of the CD47 “don’t eat me signal” which physiologically binds to signal regulatory protein α (SIRPα) on macrophages and inhibits phagocytosis of tumor cells. With an anti-CD47 antibody, phagocytosis of ATC cell lines by macrophages could be induced in vitro and in a xenotransplant model [49].

In ATC, this high macrophage infiltration in ATC tumor samples results in an altered-immunosuppressed immune microenvironment in 50% of cases and a hot immune environment in 34%, with a high expression of several inhibitory immune checkpoint mediators such as anti-cytotoxic T-lymphocyte-associated protein 4 (CTLA-4), programmed death-ligand 1–2 (PD-L1/PD-L2), TIGIT, etc., known to inhibit cytotoxic CD8+ T-cell functions [50]. Among these inhibitory immune checkpoint molecules, PD-L1 expression has been identified in 70% of ATC samples in the pre-clinical study by Schürch et al. [49] and in the phase I study of spartalizumab [10]. PD-L1 expression based on the proportion of stained tumor cells according to Tumor Proportion Score (TPS) has been found at ≥5% in 73% of 93 samples in a recent multicenter study from Germany [51]. In the tumor microenvironment, PD-L1 can be upregulated in both tumor cells and immune-microenvironment cells, such as TAMs in ATC samples [52,53]. This dual expression of PD-L1 may have important pathophysiological implications, because although induction of PD-L1 on tumor cells is interferon gamma (IFNg)-dependent and transient, PD-L1 induction on TAMs is of greater magnitude, only partially IFNg dependent and more stable over time, and thus may account for the immunosuppressive microenvironment in ATC [54]. This PD-L1 expression on TAMs may account for the presence of exhausted T-cells in transcriptomic analysis of ATC samples [50]. Moreover, PD-L1 expression on tumor cells, although possibly predictive of a response to immunotherapy in ATC [10], can be induced by the immune microenvironment, especially T-cells and TAMs, by different signaling pathways, and thus may result in differential responses to treatment with immune checkpoint inhibitors (ICI). This PD-L1 induction through different pathways by different immune cells of the tumor microenvironment may result in either accelerated tumor growth and resistance to doxorubicin and ICI of PD-L1+ tumor cells induced by TAMs; or delayed tumor growth and greater sensitivity to both doxorubicin and ICI when induced by T-cells, as shown in a hepatocellular cell line and a mouse model of hepatoma [55]. Therefore, although PD-L1 expression is high in ATC samples, its clinical impact and the differential expression on either tumor cells and/or TAMs still remains to be refined.

## 4. Treatment: To Treat Aggressively or to Palliate the Symptoms?

### 4.1. Multimodal Therapy or Palliative Care within a Fast-Dedicated Management Track

In ATC, goals of care may be therapeutic and/or palliative depending on staging and prognosis when considered in the context of available therapies, comorbidities and the patient’s wishes. Multimodal therapy refers to the combination of excisional surgery, when possible, external beam radiation therapy (EBRT), chemotherapy and/or targeted therapy (Figure 2, Figure 3 and Figure 4). This multimodal strategy is associated with a better OS in retrospective studies. In 1990, the mean OS was about two to six months [56,57] and it seems to be nearly the same 20 years later, with a one-year OS of less than 20% [58,59,60,61]. However, in a cohort of 479 patients treated in the same institution spanning nearly 20 years, Maniakas et al. found one- and two-year OS of 35/18% in the 2000/13 era (*n* = 227), 47/25% in the 2014/16 era (*n* = 100) and 59/42% in the 2017/19 era (*n* = 152) which suggests an impact of multi-modal treatment strategies on survival [14].

Dedicated fast-track management of these patients may offer a better chance of tumor control, as early management is key according to good practice guidance for fast-growing cancers [2,62]. Although feasible in highly specialized tertiary centers, the applicability of this fast-track management approach in multiple tertiary centers at a whole-country level remains to be demonstrated, as exemplified by the huge differences in reported survival between nationwide ATC database analysis [4,5,16] in comparison with tertiary-center database analysis [14,62,63,64].

Although effective in providing better OS, multimodal treatment is, in most cases, a palliative aggressive approach with risks of side effects and complications which may hamper the quality of life of the patients. Indeed, in large national series, the proportion of patients unfit for combined treatments varies from 4% (*n* = 4/100) [5] to 15% in the French network on refractory thyroid cancer (ENDOCAN-TUTHYREF) experience (unpublished data). Therefore, a review of the treatment options, risks, benefits and outcomes has to be submitted to a multidisciplinary team (including palliative caregivers and geriatric oncologist) and presented to the patient to create a shared decision-making process about a realistic treatment plan [2]. One critical issue is to clarify with the patient and his family whether tracheostomy should be performed in case of acute respiratory failure versus palliative sedation, because such a procedure would profoundly impact their ability to communicate and their quality of life until death.

Whenever possible, surgery must be performed as it can provide prolonged survival and even a cure in the 10% patients with stage IVa disease, in association with chemo-radiotherapy and surgery [63,65]. It should also be performed in patients with advanced disease who may respond to initial medical/radiation treatment [2,14] (Figure 2 and Figure 3).

### 4.2. Radiation Therapy: Still the Mainstem of ATC Treatment

The aggressive nature of the disease results in a high rate of local progression and recurrence [66,67] which require achieving local control with surgery, when feasible, and, more often, with EBRT. There has been a high heterogeneity of EBRT reports in the last 25 years in terms of dose administration, fractionation, techniques and combinations [26].

Since the first result from a retrospective study by Aldinger et al. in 1978 [26], and despite the absence of prospective trials, EBRT is recommended because it has been shown to improve median OS in retrospective studies, including reports from large nationwide databases such as SEER [5,19,68,69,70,71,72,73,74,75,76]. Moreover, this improved prognosis with EBRT is obtained through a multimodal treatment as shown, for example, in the SEER database analysis by Song et al. which reported EBRT, surgery and chemotherapy as prognostic factors on OS in multivariate analysis in 433 stage IVc patients (hazard ratio (HR): 0.562, *p* ≤ 0.001) [76].

In order to provide the best survival benefit to the patients, EBRT dose matters. Indeed, the total dose delivered seems to be predictive of survival and local control in most of the studies, with 45–60 Gy providing an optimal control, whatever the type of fractionation, across different studies [5,59,60,77,78]. In a report of 31 patients with no distant disease at presentation who were treated by chemoradiotherapy (2 Gy daily fraction) +/− surgery, a total dose >50 Gy was associated with a significantly better median OS (9.3 vs. 1.6 months, *p* = 0.019) [79]. In the National Cancer Database (NCDB) analysis on 1288 patients with non-resected ATC, Pezzi et al. reported an improved one-year OS rate for Stage IVb and IVc patients who received 60 to 75 Gy as compared with patients treated with less than 60 Gy (31% versus 16%, *p* = 0.019) [73]. To better define the optimal dose, Nachalon et al. analyzed the total EBRT dose to the gross tumor in three different groups of 26 patients, using full dose (70 Gy), high palliative dose (50 Gy) and palliative dose (maximum 30 Gy), and found an association between dose and improved survival (*p* < 0.001) [80]. Following R0 or R1 resection, guidelines recommend that good-performance status patients with no evidence of metastatic disease who opt for aggressive management should be offered standard fractionation IMRT with concurrent systemic therapy [2].

Besides dose, fractionation modalities have been investigated for optimal tumor control in ATC patients. Bi-fractionation radiotherapy schedules can be either hyper-fractionated (multiple daily doses not exceeding 1.5 Gy) or accelerated (twice-a-day dose of 1.8–2 Gy). These schedules have been used in an attempt to overcome fast progression and radioresistance of ATC [81]. However, there have been no randomized studies to compare standard with altered fractionation; such studies may not be feasible given the rarity of this disease. In 2002, Tennvall et al. published prospective results for patients treated by association of chemotherapy and hyper-fractionated EBRT (30 Gy before surgery and 16 Gy after, twice daily either as 2 × 1 Gy or 2 × 1.3 Gy per day or 46 Gy after, twice daily as 2 × 1.6 Gy) and debulking surgery. They found a trend for hyper-fractionated radiotherapy with a better OS outcome (9% of patients (5/55) always alive after two years) and 60% (33/55) of local control [82]. In 2006, Wang Y et al. reported results from 23 ATC patients treated with radical EBRT (dose > 40 Gy), with once-daily (*n* = 14) or twice-daily fractionation (*n* = 9, 1.5 Gy per fraction). They found that hyper-fractionated treatment was well tolerated with a longer median OS (13.6 months for twice-daily radiotherapy versus 1.3 months for the other group; however, this was without statistical significance (*p* = 0.3). There were also no difference in terms of local control in the two groups [83]. A prospective study using accelerated radiotherapy with two daily fractions by De Crevoisier et al. provided encouraging results with 7 patients out of 30 alive after a median follow-up of 45 months and 47% local control rate [63]. Although bi-fractionation may provide better local tumor control, there is no definitive evidence and no data to suggest an improved OS with these techniques. Data on ATC are scarce but we can extrapolate from head and neck cancer data where altered fractionation increases acute but not late toxicity and improves local control [84].

Hypofractionation with dose per fraction > 5 Gy has been reported to prevent death from local recurrence (*p* = 0.025) but with grade > 3 toxicities in a series of 33 patients and no improved OS [85]. However, because hyperfractionation or accelerated regimens require more intensive resources and the data supporting their efficacy are scarce, recent studies have used regimens close to those used in EBRT of other head and neck cancers in 2 Gy per fraction [60,78,86,87,88,89]. In recent guidelines from the Italian Society of Radiation Oncology (AIRO) and Spanish Thyroid Group, bifractionation and standard fractionations are listed as two possible EBRT options, whereas hyperfractionation is discouraged in ESMO guidelines [3,90,91]. ATA guidelines recommend standard fractionation but do not discourage hyperfractionation [2].

Although different fractionation techniques do not seem to have a major impact in management, high-accuracy radiotherapy techniques such as intensity-modulated radiation therapy (IMRT) should also be used for decreasing toxicities [92,93]. Indeed, IMRT can achieve a better target coverage and reduced dose to the spinal cord [92]. In a monocentric retrospective analysis of 28 ATC patients treated with IMRT and 13 treated with three-dimensional conformal radiotherapy (3D-CRT), a better progression-free survival (PFS) (median 5.1 vs. 2.6 months, *p* = 0.049) and OS (*p* = 0.005 for both) were found in multivariate analysis, possibly related to an improved total dose (*p* = 0.005) without increased toxicity [94]. This lower toxicity rate was not found in all reports of IMRT [79]. This IMRT technique is important to deliver the optimal dose to the different treatment volumes which most often include the operative bed, the thyroid, and lymph node areas I to VI with the upper mediastinum up to the carina [63,78,79,89,95]. Few relapses at the edge of the radiotherapy field (less than 10% marginal relapses) have been reported [96,97]. The rate of loco-regional failure rate may relate to the extent of the EBRT field, as reported by Kim et al., with a five-year local control rate of 40% in ATC patients treated with a large extended field (*n* = 12) (such as described before) vs. 9% in case of limited field EBRT (*n* = 11) to involved site (tumor and node) (*p* = 0.04). [98]. We proposed EBRT by IMRT with a total dose of 66–70 Gy into the tumor volume with a 5 mm margin and 50 Gy to the bilateral level II–VI cervical nodes, and upper mediastinal nodes to the carina +/− area I, with an element of compromise therefore required in efforts to achieve acceptable toxicity.

The optimal timing of EBRT in the patient’s treatment schedule with respect to surgery and chemotherapy/systemic treatment has not been defined. In patients with resectable stage IVa disease, adjuvant EBRT is recommended [2]. However, it is possible that preoperative radiotherapy can enable surgery (example in Figure 2). In a monocentric report of 79 patients treated between 1972 and 1998, Besic et al. found that the 12 patients treated by neoadjuvant chemo-radiotherapy, surgery and additional adjuvant EBRT (total dose of 45 to 64 Gy) had a better median OS (14.5 months as compared with the 26 patients treated with primary surgery followed by EBRT (7 months)) [99]. Although interesting, these results may reflect a selection bias with neoadjuvant treatment allowing for the selection of patients without primary refractoriness and thus, better prognosis. Indeed, Arora et al. found a longer cause-specific survival in cases of postoperative EBRT versus preoperative EBRT (*p* < 0.0001) in an analysis of PDTC and ATC patients from the SEER database [100]. Data are still needed to confirm the best timing for EBRT in patients with resectable disease.

Beside surgery, the other major question is the optimal systemic treatment to combine with EBRT and especially chemotherapy. Since the 1970s, several authors have described the results of combinations, and there are a large number of different protocols. One of the most recent analyses of the SEER database, which included an inverse probability weighting (IPW) to balance variables between groups, reported that radio-chemotherapy was associated with improved OS (adjusted HR = 0.69, 95% CI: 0.56–0.85, *p* < 0.001) versus EBRT alone; this difference remained significant within each subgroup stratified by surgical resection and distant metastasis [101]. The most recent and only randomized study in this setting is the RTOG 0912 trial which randomized 99 patients (56% Stage IVb) undergoing 2–3 weeks of weekly paclitaxel 80 mg/m^2^ with placebo or pazopanib 400 mg/day followed by concomitant 66 Gy EBRT and weekly 50 mg/m^2^ paclitaxel with placebo or pazopanib 300 mg/day [102]. The one-year OS rate was 29% in the placebo group and 37.1% in the pazopanib group (*p* = 0.283) in the 80% of patients who were randomized and eligible for this treatment. This randomized trial provides the prospective results of chemo-radiation therapy in ATC patients which are in line with those of the different retrospective studies which reported one-year OS rates between 30 and 50% with either doxorubicin or paclitaxel/docetaxel-based chemotherapy, with or without platin [63,65,74,103] (Table 1). No chemotherapy regimen has been shown to have a better impact on OS in the different retrospective studies published to date (Table 1). In the late 1990s and early 2000s, bi-fractionated radiotherapy (46 Gy, 1.6 Gy per fraction) associated with doxorubicin (10–20 mg/m^2^ weekly) was widely considered as the standard of care but at least 10 different chemotherapy regimens were found in different series or even in single-institution cohorts [57,67,79] (Table 1). De Crevoisier et al. have studied the association of doxorubicin (60 mg/m^2^) and cisplatin (120 mg/m^2^) with radiotherapy (40 Gy in 1.23 Gy per fraction twice a day) within a prospective cohort. They reported a three-year OS of 27% (95% CI: 10–44%) and a median OS of 10 months [63]. In a German multicenter study, Wendler et al. found that any kind of systemic treatment (doxorubicin weekly, paclitaxel, paclitaxel and pemetrexed, paclitaxel and carboplatin, doxorubicin and cisplatin and tyrosin kinase inhibitor) was associated with a longer OS for IVc patients (HR: 0.23, 95% CI: 0.08–0.64, *p* = 0.005), and combined paclitaxel and pemetrexed was associated with a statistically significant likelihood of longer OS compared with other regimens (*p* < 0.0001) [5]. To summarize, most guidelines recommend, in eligible patients without targetable molecular alteration, the administration of chemo-radiation with either doxorubicin or paclitaxel/docetaxel +/− platin. This chemo-radiotherapy constitutes the standard induction treatment for fit stage IVb-IVc patients and has been referred to as “bridging therapy” in the recent ATA guidelines: a therapy that helps to stop the fast progression of the disease while all molecular explorations are undertaken and accessibility to targeted therapy/immunotherapy is evaluated [2]. Finally, for patients with *BRAF* mutations who are eligible for the dabrafenib/trametinib combination, the question of when to perform radiation therapy remains open as it is usually indicated to halt treatment during EBRT. However, a phase I/II trial in melanoma has shown that concurrent radiation could be feasible [104] and a phase I trial in ATC patients to evaluate the feasibility of concurrent EBRT and dabrafenib/trametinib combination (NCT03975231) is ongoing (Table 2).

### 4.3. Systemic Therapies: Failure of Chemotherapies, Success of Targeted Therapies and the Promises of Immunotherapy

Chemotherapy and EBRT are independent prognostic factors that are associated with improved survival [4,5,6,14,16,117,118]. However, clinical trials comparing different chemotherapy regimens in ATC patients are scarce because of the disease’s low incidence and aggressiveness limiting enrollment in clinical trials, leading to poor statistical power and a limited treatment time-frame. In the absence of molecular abnormalities, the most recent ATA Guidelines recommend starting with systemic therapy with genotoxic drugs such as paclitaxel and carboplatin combinations, cisplatin and doxorubicin combinations, docetaxel and doxorubicin combinations, paclitaxel alone, or doxorubicin alone [2,3,119,120]. Table 1 summarizes the studies on the different chemotherapy protocols in ATCs. Given the estimated doubling time of ATC is only 3–12 days, the interval between the administrations of the chemotherapeutic agent has to be short [63]. In this objective, some authors recommend using chemotherapeutic regimens at relatively short intervals (such as weekly administration compared with 3–4-week intervals). The poor prognosis of ATCs is often associated with primary chemoresistance which results in an average PFS of less than three months (Table 2). Indeed, though paclitaxel seems the most effective chemotherapeutic drug, chemo-resistance is common, which may be related to TAMs infiltration. TAMs occupy 50% of the tumor volume and provides paracrine signals via the CSF-1/CSF-1R axis, which promotes tumor progression and therapy resistance. Thus, targeting the CSF-1/CSF-1R pathway in TAMs was shown to restore the sensitivity of thyroid cancer cells to paclitaxel [121].

Due to this primary chemoresistance, other therapeutic strategies have been developed. High-throughput sequencing investigations have unveiled the molecular alterations of ATC opening the way to targeted therapies [7,14,28,29,61]. The recent approval of a combination therapy with the BRAF inhibitor dabrafenib and the MEK inhibitor trametinib for patients with unresectable or metastatic *BRAF^V600E^*-positive ATC has generated enthusiasm in the field. Indeed, a phase II basket trial describing the efficacy and safety of dabrafenib plus trametinib has enrolled 36 ATC patients, the median age of whom was 71 years; 30/36 (83%) patients had undergone prior tumor radiation. The ORR was 56% (95% CI: 38.1–72.1%), including three complete responses and a median PFS and OS of 6.7 and 14.5 months, respectively, without new safety signals identified with this additional follow-up confirming the results of previous studies [8,9,122]. These results have led to approval by the FDA but not by the EMA. Unfortunately, this combination can only be offered to the 20–50% of ATC patients with *BRAF^V600E^* mutation and acquired resistance to BRAF inhibitors may develop via secondary mutations in the MAPK pathway or via the PI3K/AKT/mTOR pathway, or Hgf/Met activation, underlining the need for additional rationally designed approaches [123,124,125]. Much rarer than *BRAF^V600E^* mutations are *NTRK, ALK* and *RET* fusions which can be found in 2–3% of ATC patients. However, finding one of these alterations might greatly impact the prognosis of the few mutated patients given the very high response rates observed with their corresponding highly specific inhibitors in different basket trials. A pooled subgroup analysis of the larotrectinib trials (NCT02122913 and NCT02576431) reported a 29% response rate in two out of seven ATC patients; responses lasted for 3.7 and 10.2 months and the median OS was 14.1 months (95% CI: 2.6–NE) [126]. A long-lasting, dramatic response to the RET-specific inhibitor selpercatinib has been reported in a 73 year old patient with *CCD6-RET* fusion ATC [127]. Lasting responses to ALK inhibitor in patients with ALK fusions have also been reported [128]. Although these targeted therapies have demonstrated impressive activity, availability is a major issue in most countries except in the US, where approval has been obtained from the FDA. On 31 July 2020, a conditional marketing authorization valid throughout the European Union (EU) was issued for entrectinib for the treatment of adults with NTRK fusion-positive solid tumors that are locally advanced, metastatic or where surgical resection is likely to result in severe morbidity. To the best of our knowledge, no approvals have been obtained in ATC for RET inhibitors. All protocol regimens are presented in Table 3.

Inhibitors targeting the PI3K/AKT/mTOR pathway, such as everolimus, have been tested but the results were disappointing, with none of the seven patients included in a phase II study benefiting from treatment [129]. A multicenter, phase II trial of everolimus in locally advanced or metastatic thyroid cancer of all histologic subtypes included six ATC cases with only one response, and the others had progressive disease with a median PFS of 10 weeks [130]. Another phase II study evaluated the combination of sorafenib and temsirolimus in two patients with ATC, of which only one had an objective response [131]. To our knowledge, no trials have looked at mTOR inhibitors in the context of ATC with mutation in the PI3K/AKT/mTOR pathway. Antiangiogenic treatments such as lenvatinib [132], a multikinase inhibitor approved in differentiated thyroid cancers, have shown initially encouraging results but limited activity in subsequent studies, with a risk of bleeding and fistula as this disease often invades the trachea, esophagus and vessels, and are thus not recommended [2,77,133]. A recent prospective phase II trial was halted for futility as the minimum ORR threshold of 15% was not met upon interim analysis with a 2.9% response rate, a median PFS of 2.6 months and a median OS of 3.2 months [134].

Over the past few years, immuno-oncologic treatments, especially ICI (e.g., anti-PD-1, anti-PD-L1 and anti-CTLA-4) have revolutionized the field of anti-cancer therapies in many entities. PD-L1 has been suggested as a predictive biomarker of response to ICI in several cancers although its robustness has been questioned. As described above, PD-L1 is often expressed on ATC tumor cells, suggesting new treatment opportunities for ATC with immunotherapy [10,51]. Indeed, spartalizumab, an anti-PD-1 antibody, has been studied in ATC [10]. The response rate was 19% (five PR and three CR observed). The median OS in the entire cohort was 5.9 months, with 40% of patients alive at one year. The median PFS was 1.7 months. Interestingly, those patients with PD-L1 expression of <1% had a median OS of 1.6 months and there were no responses in this group; however, those with PD-L1 expression of 1–49% and ≥50% had a median OS that had not been reached and an overall response rate of 18% (2/11) and 35% (6/17), respectively. The highest rate of response was observed in the subset of patients with PD-L1 > 50% (6/17; 35%). It should be noted that spartalizumab is not FDA- or EMA-approved and is not commercially available. The ACSé basket trial evaluated pembrolizumab in rare cancers in France and included a cohort of 16 ATC patients. The response rate was 25% with a median duration of response of 7.3 months in responder patients [13]. These results differ from those of a phase II study of pembrolizumab combined with chemoradiotherapy as initial treatment which enrolled only three patients, because all three patients died within six months [53]. One might hypothesize that ICI would be more effective without concurrent radiation therapy but further data will be needed to evaluate the best timing for ICI initiation and its clinical efficacy, and clinical trials are ongoing (Table 2). Although effective, ICI will only benefit a small group of patients. Therefore, combination strategies have been developed. In a phase II umbrella study, anti PD-L1 atezolizumab has been studied in combination with either vemurafenib or cobimetinib for *BRAF*-mutated patients (cohort 1), cobimetinib alone for *RAS-* and *NF1*-mutated patients (cohort 2) and bevacizumab for patients with no mutation (cohort 3) in a prospective multi-arm trial [135]. Median OSs were not reached in cohort 1; these were 18.23 months in cohort 2 and 6.21 months in cohort 3. The response rate was 71% in cohort 1 and 7% in cohort 2. Therefore, combination of ICI with targeted therapies in ATC patients with molecular alterations is very promising. In a retrospective study, Diercks et al. analyzed six patients with metastatic ATC treated with multikinase inhibitors (lenvatinib) and ICI (pembrolizumab) and showed 66% with complete remissions (4/6), 16% with stable disease (1/6), and 16% with progressive disease (1/6). The median PFS was 16.8 months and the median OS was 17.3 months [11].

These optimistic data may lead to a systematic screening of PD-L1 and/or MMR status. However, it will first be necessary to define a specific expression score for PD-L1 expression in ATCs and to correlate it with the clinical benefit. Because the ATC immune microenvironment is an immunosuppressive medium, the development of immunotherapy combinations to improve these results will also be required. Clinical trials using immunotherapy in combination with other systemic agents are underway (Table 2).

### 4.4. Reappraisal of Surgery in the Era of Targeted Therapies

There has been substantial debate about the appropriate role for surgery in the management of locally advanced and metastatic ATC. Operative treatment of local disease offers the best opportunity for prolonged survival if the neoplasm is intrathyroidal. When the neoplasm is extrathyroidal, the operative approach is controversial, as some have found that neither the extent of the operation nor the completeness of the tumor resection affects survival [63,77,136,137]. Complete resection is recommended whenever possible for patients with confined ATC (stage IVa/IVb) in whom R0/R1 resection is anticipated, if excessive morbidity can be avoided [2,3,138]. Lateral compartment lymphadenectomy should be performed only in the setting of complete macroscopic resection. Resection of the larynx, pharynx and esophagus are discouraged [56,67,139].

ATC patients present with extensively invasive primary tumors in between 85 and 95% of cases [68,140] justifying the use of neoadjuvant therapy by chemotherapy, chemo-radiotherapy (Figure 3) or by targeted therapy [82,99,115,141]. In BRAF^V600E^-mutated ATC patients, the successful use of neoadjuvant dabrafenib plus trametinib with or without immunotherapy has been described in a case series of six patients that eventually underwent surgery [142,143]. In these patients, adjuvant therapy was prescribed with the *BRAF-*directed therapy or with chemoradiation after surgery. OS at 6 and 12 months was 100% and 83%, respectively, but the locoregional control rate was 100%. These data have revigorated interest toward primary resection upfront or following initial response to treatment in ATC patients.

## 5. Conclusions

ATCs are aggressive tumors occurring most often in the elderly. The survival outcomes of anaplastic thyroid carcinoma remain poor. Their diagnostic and therapeutic management must be initiated quickly and has to be coordinated within an expert center network. The objectives of the treatment are to fight against the risk of suffocation, to control the tumor mass and to ensure optimal treatment of symptoms within a multidisciplinary team involving endocrinologists, medical oncologists and radiotherapists, palliative care, geriatric oncologists, surgeons, and radiologists in order to offer appropriate care at each stage of the disease. Multimodal treatment combining surgery, radiotherapy, chemotherapy or targeted therapy can allow the control of the tumor. The results of immunotherapy are encouraging but its place is yet to be defined. With increasing knowledge of the tumor biology, the identification of the underlying molecular pathways, modifications of the transcriptome, proteome and associated immunomodulatory mechanisms of ATC (TAMs) on the one hand, and the emerging role of novel, molecular-based single/multi-targeted therapies on the other hand, a growing number of clinical trials can be noted. Prioritizing clinical trial enrollment will be a key factor in advancing care for patients with ATC.

## Figures and Tables

**Figure 1 cancers-14-01061-f001:**
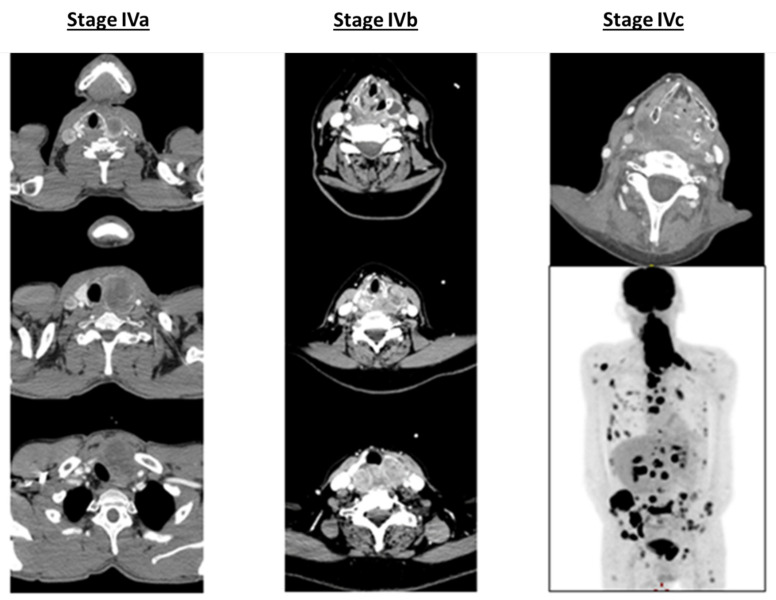
CT scan illustration of anaplastic thyroid carcinoma (ATC) staging.

**Figure 2 cancers-14-01061-f002:**
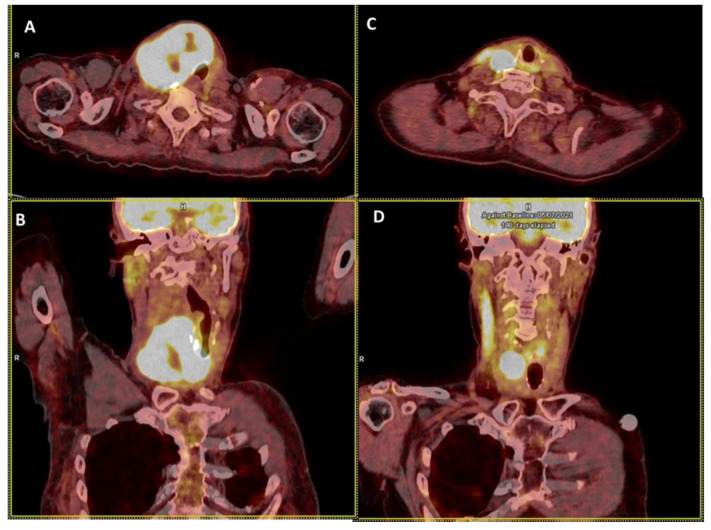
Illustration of ^18^FDG PET/CT anaplastic thyroid carcinoma (ATC) response to chemotherapy and radiotherapy before surgery. The figure represents axial (**A**) and coronal (**B**) slices of the neck region with an ATC volume of 230 mL before initiating treatment comprising chemotherapy (Cisplatin-Doxorubicin, 2 cures) and radiotherapy (IMRT, 50 Gy. (**C**,**D**) represent axial (**C**) and coronal slices (**D**) of the neck region with an ATC volume of 15 mL.

**Figure 3 cancers-14-01061-f003:**
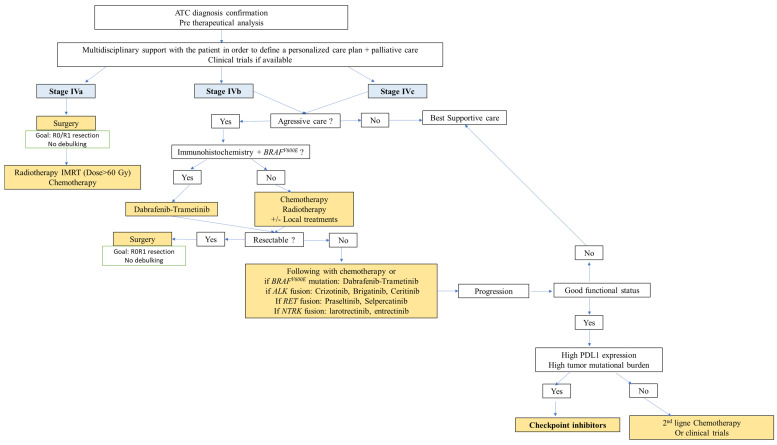
Initial treatment of stages IVa, IVb and IVc ATC, adapted from [2].

**Figure 4 cancers-14-01061-f004:**
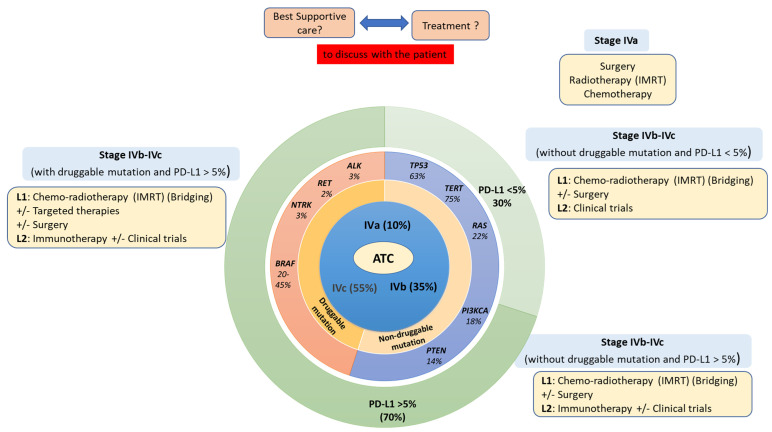
Molecular and treatment landscape of ATC (based on [2]). ATC: Anaplastic Thyroid Carcinoma, L1 and L2: Line of treatment one or two, IMRT: Intensity-Modulated Radiation Therapy, PD-L1: Programmed Death-Ligand 1.

**Table 1 cancers-14-01061-t001:** Outcome of multimodal treatment in ATC.

Authors, References	Study	Number of Patients (Total and According to the Stage	Surgery	Radiotherapy (Dose)	Chemotherapy Protocol (*n* = Number of Patients)	Outcomes (ORR (*n* and %), Median OS (Months) and PFS (Months) Local and Distant Control at 1 Year (%))
[105]	Retrospective	*n* = 19 metastatic: *n* = 9	12	30	Bleomycin + Cyclophosphamide + 5-FU	OS: 7–12 months ORR, PFS, local control: ND Distant control: 16%
[106]	Prospective, randomized,	*n* = 39	ND	ND	Doxorubicin (*n* = 21) Doxorubicin + Cisplatin (*n* = 18)	ORR: 1/21 (4.8%); OS: ND; PFS: ND ORR: 6/18 (22.2%); OS: ND; PFS: ND
[107]	Retrospective	*n* = 34	15	30	Bleomycin + Cyclophosphamide + 5-FU	ORR: 22/34 (64%); OS: 4 months; PFS: ND
[108]	Prospective, no randomization, non-controlled	*n* = 19 (limited to the neck)	10	19 (57.6 Gy)	Doxorubicin per week (*n* = 19)	OS: 12 months Local control: 68% Distant control: 21%
[109]	Prospective randomized but non-controlled	*n* = 20 metastatic: *n* = 6 metastatic: *n* = 3	12 −9 −3	20	Doxorubicin + Cisplatin (*n* = 12) Mitoxantrone (*n* = 8)	OS: 2–6 months; PFS: 6.5 months Local control: 9/12 (75%); Distant control: 4/12 (33.3%); Local control: 3/8 (37.5%); Distant control: 0/12 (0%); PFS: 4 months
[110]	Prospective, no randomization, non-controlled	*n* = 32 Stage IVb: *n* = 23 Stage IVc: *n* = 9	0/32	32 (30–45 Gy)	Doxorubicin (*n* = 14/32)	OS: 6 months; PFS: ND Local control: 7/32 (21.9%); Distant control: ND
[111]	Retrospective	*n* = 89	10	ND	Vinblastine or Cisplatin or Doxorubicin or Novantrone	ND
[56]	Retrospective	*n* = 121 Stage IVc: *n* = 64	106	58	*n* = 64	OS: 6 months (mean OS: 7.2 +/− 10 months) ORR, PFS, local or distant control: ND
[21]	Retrospective	*n* = 17	17	12	ND	OS: 12 months
[82]	Prospective study, randomized but non-controlled	*n* = 55 Stage IVc: *n* = 17	40	55 (46 Gy)	Doxorubicin (*n* = 55)	OS: 2–4.5 months ORR local: 60% ORR distant: 22% PFS: ND
[63]	Prospective, no randomization, non-controlled	*n* = 30 Stage IVa: *n* = 4 Stage IVb: *n* = 20 Stage IVc: *n* = 6	7	30 (40 Gy)	Doxorubicin + Cisplatin	OS: 10 months Local control: 47%; Distant control: 37%
[60]	Retrospective	*n* = 37	19	37 (57.6 Gy)	Doxorubicin (*n* = 37)	OS: 6 months Median Loco-regional-PFS: 10.1 months Local control: 45%
[112]	Retrospective	*n* = 44 Local: *n* = 12 Regional: *n* = 12 Distant: *n* = 20	44	39 (46–50 Gy)	Doxorubicin + Cisplatin (*n* = 33) Doxorubicin + Carboplatin (*n* = 3) Doxorubicin (*n* = 1) Paclitaxel (*n* = 1)	OS: 8.5 months ORR: 22/44 (50%) PFS: 6.5 months
[113]	Retrospective	*n* = 13 Stage IVc: *n* = 6	8	5 (45–65 Gy)	Doxorubicin (*n* = 5)	OS: 3.8 months ORR: ND PFS: 2.8 months
[6]	Retrospective	*n* = 547 Stage IVa: *n* = 69 Stage IVb: *n* = 242 Stage IVc: *n* = 233	n = 301	319	*n* = 255 Etoposide + Cisplatin (EP) Etoposide + Cisplatin + Doxorubicin 5FU + Cisplatin + Doxorubicin Paclitaxel	Stage-dependent OS: Stage IVa: 7.8 months Stage IVb: 4.8 months Stage IVc: 2.7 months PFS/ORR: ND
[114]	Phase 3	*n* = 80 Stage IVa: *n* = 1 Stage IVb: *n* = 6 Stage IVc: *n* = 72 ND: *n* = 1	44/80	*n* = 28	Fosbretabulin + Carboplatin + Paclitaxel Control: Carboplatin + Paclitaxel	OS: 8.2 months if surgery versus 4.0 months on the control arm OS: 4.0 months if no surgery and 4.6 months on the control arm
[115]	Prospective, controlled, non-randomized	*n* = 13 Stage IVb, *n* = 9 Stage IVc, *n* = 4	4	ND	Paclitaxel	OS and PFS: ND ORR Stage IVb: 33% ORR Stage IVc: 25%
[64]	Retrospective	*n* = 92 Stage IVa: *n* = 6 Stage IVb, *n* = 22 Stage IVc, *n* = 61 ND: *n* = 3	35	56 (55 Gy)	59 Doxorubicin + Cisplatin (*n* = 56) Carboplatin + Paclitaxel (*n* = 3)	OS: 7 months; PFS: 5 months Local control: 75%; Distant control: 63%
[116]	Retrospective	*n* = 8 Stage IVb: *n* = 2 Stage IVc: *n* = 4 ND: *n* = 2	6	5 (40–60 Gy)	Docetaxel + Cisplatin	OS: 30.4 months ORR: 3/8 (37.5%) PFS: 5.5 months
[73]	Retrospective	*n* = 1288 Stage IVc: *n* = 608	0	613	471 (treatment not available)	OS: 2.27 months
[5]	Retrospective	*n* = 100 Stage IVa: *n* = 9 Stage IVb: *n* = 32 Stage IVc: *n* = 54 ND: *n* = 5	83	81 (57.6 Gy)	Doxorubicin weekly (*n* = 25) Paclitaxel weekly *(n* = 9) Paclitaxel + Pemetrexed (*n* = 8) Doxorubicin + Cisplatin (*n* = 8) Carboplatin + Paclitaxel (*n* = 14) Tyrosine kinase inhibitors (*n* = 10) Other (*n* = 10)	Median OS: 5.7 months Stage-dependent OS (months and % at 1 year): Stage IVa: 26 months (66%) Stage IVb: 11 months (39%) Stage IVc: 3 months (13%) ORR and PFS: ND
[74]	Retrospective	*n* = 30 Stage IVa: *n* = 2 Stage IVb: *n* = 22 Stage IVc: *n* = 6 ND: *n* = 5	27	30 (66 Gy)	Doxorubicin + Docetaxel (*n* = 19) Carboplatin + Paclitaxel (*n* = 5) Doxorubicin only (*n* = 4) Cisplatin only (*n* = 2)	Median OS: 21 months Median PFS: 8.3 months ORR: 19/30 (63.3%) Local control: 93% Distant control: 22%
[65]	Retrospective	*n* = 44 Stage IVa: *n* = 10 Stage IVb: *n* = 17 Stage IVc: *n* = 27	23	29	platinum or taxane based agents (*n* = 46)	OS: 11.9 months (total cohort) and 22.1 months in patients treated with chemotherapy and EBRT TTF: 3.8 months

OS: Overall survival, PFS: Progression-free survival, ORR: Objective response rate, ND: Non-determinated, TTF: Time to treatment failure, EBRT: External beam radiation, ATC: Anaplastic thyroid carcinoma, IMRT: Intensity-Modulated Radiation Therapy, NA: non-applicable. Advanced ATCs, understanding advanced ATCs locally and/or at distant sites.

**Table 2 cancers-14-01061-t002:** Ongoing clinical trials in ATC patients.

Clinical Trials Gov. Identifier	Treatments/Interventions (Settings)	Phase	Status
NCT03565536	Sorafenib (Neoadjuvant treatment of ATC)	Phase 2	Unknown
NCT03085056	Trametinib + Paclitaxel (Advanced ATC)	Early Phase 1	Recruiting
NCT02688608	Pembrolizumab (Advanced ATC)	Phase 2	Unknown
NCT02244463	MLN0128 (Advanced ATC)	Phase 2	Active, not recruiting
NCT04739566	Dabrafenib + Trametinib (Neoadjuvant Strategy in ATC with *BRAF* mutation)	Phase 2	Recruiting
NCT03122496	Durvalumab + Tremelimumab + Stereotactic Body Radiotherapy (Advanced ATC)	Phase 1	Active, not recruiting
NCT01236547	IMRT + Paclitaxel with or without Pazopanib Hydrochloride (Advanced ATC)	Phase 2	Active, not recruiting
NCT05102292	HLX208 (Advanced ATC with *BRAF^V600^* mutation)	Phase 1b/2	Recruiting
NCT02152137	Efatutazone + Paclitaxel (Advanced ATC)	Phase 2	Active, not recruiting
NCT04552769	Abemaciclib (CDK4 + CDK6 inhibitor) (Advanced ATC)	Phase 2	Recruiting
NCT04675710	Pembrolizumab + Dabrafenib + Trametinib (Neoadjuvant *BRAF-*Mutated ATC)	Phase 2	Recruiting
NCT04238624	Cemiplimab + Dabrafenib + Trametinib (Advanced ATC)	Phase 2	Recruiting
NCT04420754	AIC100 Chimeric Antigen Receptor T-cells (Relapsed/Refractory Thyroid Cancer)	Phase 1	Recruiting
NCT03975231	Dabrafenib + Trametinib + IMRT in (Advanced *BRAF* Mutated ATC)	Phase 1	Recruiting
NCT03449108	LN-145/LN-145-S1 (Autologous Centrally Manufactured Tumor Infiltrating Lymphocytes) (Advanced ATC)	Phase 2	Recruiting
NCT04592484	CDK-002 (exoSTING) (Advanced/Metastatic, Recurrent, Injectable ATC)	Phase 1	Recruiting
NCT03181100	Cohort I (*BRAF* mutation): Vemurafenib + Cobimetinib + Atezolizumab. Cohort II (*RAS*, *NF1* or *NF2* mutations): Cobimetinib + Atezolizumab Cohort III (non *BRAF* or *RAS* mutation): Bevacizumab + Atezolizumab Cohort IV: Nab-paclitaxel + Atezolizumab	Phase 2	Recruiting
NCT03246958	Nivolumab + Ipilimumab (Advanced ATC)	Phase 2	Active non-recruiting
NCT04400474	Cabozantinib + Atezolizumab (Advanced ATC)	Phase 2	Recruiting
NCT04579757	Surufatinib + Tislelizumab (Advanced ATC)	Phase 1/2	Recruiting
NCT04759911	Selpercatinib (Neoadjuvant ATC with RET alterations)	Phase 2	Recruiting

ATC: Anaplastic thyroid carcinoma, IMRT: Intensity-modulated radiation therapy, NA: non-applicable. Advanced ATCs, understanding advanced ATCs locally and/or at distant sites.

**Table 3 cancers-14-01061-t003:** Treatment protocols in Anaplastic Thyroid Carcinoma.

Treatment	Protocols and Dose	
Chemotherapy	Every 3 or 4 weeks Doxorubicin (60 mg/m^2^) + Cisplatin (120 mg/m^2^) every 4 weeks Paclitaxel (175 mg/m^2^) + Carboplatin (AUC 5) every 3 weeks Docetaxel (60 mg/m^2^) + Doxorubicin (60 mg/m^2^) every 3–4 weeks Paclitaxel (135–200 mg/m^2^) every 3–4 weeks Doxorubicin (60–75 mg/m^2^) every 3 weeks Every week Paclitaxel 50–100 mg/m^2^ + Carboplatin AUC2 Docetaxel (20 mg/m^2^) + Doxorubicin (20 mg/m^2^) Paclitaxel (30–60 mg/m^2^) Docetaxel (20 mg/m^2^)	
BRAF and MEK inhibitors	Dabrafenib 150 mg twice daily + Trametinib 2 mg once daily	
RET inhibitor	Selpercatinib 160 mg twice daily, reduced to 120 mg twice daily in patients weighing less than 50 kg	
	Praseltinib 400 mg per day	
NTRK inhibitor	Larotrectinib 100 mg twice daily Entrectinib 600 mg once daily	
ALK inhibitor	Crizotinib 250 mg twice daily	
	Larotrectinib 100 mg twice daily	

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
