# Peer review of "Anaplastic Thyroid Carcinoma: An Update"

_cancers, 2022, doi:10.3390/cancers14041061_

Round 1
Reviewer 1 Report
Jannin et al. wrote a pleasant and comprehensive review, especially from a clinical point of view, of the latest advances in the management and treatment of ATC, which is easily accessible even to non-experts. I particularly appreciated the reference to epidemiological studies on the various world territories. I only have a couple of suggestions to improve the manuscript:
- The only slightly under-treated part of the manuscript is the biological one relating to the genetic events responsible for the history of the tumor. For example, no reference is made to nuclear accumulation of b-catenin due to mutations in the CTNNB1 gene that are limited to PDTCs and ATCs (see Garcia-Rostan et al. 2001 and the chapter 8 by Viglietto and De Marco in Contemporary aspects of Endocrinology edited by E. Diamanti-Kandarakis). For a balanced update on this topic, the authors should extend this part.
- Some typos are present:
Line 124: PDL-L1 (corrected form PD-L1)
Line 138: TAM (corrected form TAMs)
Line 172: PECT/TC (corrected form PET/TC)
Please check carefully. There could be more
Author Response
RE: Response to Reviewer Comment
Manuscript ID: cancers-1584919
Anaplastic thyroid carcinoma: an update
ANSWERS TO THE COMMENTS BY REVIEWER 1
Reviewer #1: Jannin et al. wrote a pleasant and comprehensive review, especially from a clinical point of view, of the latest advances in the management and treatment of ATC, which is easily accessible even to non-experts. I particularly appreciated the reference to epidemiological studies on the various world territories. I only have a couple of suggestions to improve the manuscript:
Comment 1: The only slightly under-treated part of the manuscript is the biological one relating to the genetic events responsible for the history of the tumor. For example, no reference is made to nuclear accumulation of b-catenin due to mutations in the CTNNB1 gene that are limited to PDTCs and ATCs (see Garcia-Rostan et al. 2001 and the). For a balanced update on this topic, the authors should extend this part.
We have modified the text in line with your comment:
“ If regulation of cell cycle has a crucial role in oncogenesis and particularly in ATC, also protein me-tabolism control has been deeplyis also involved in tumorigenesis. For example, about 10% of patient with ATC harbor EIF1AX mutations, which has recently been involved in deregulating protein synthe-sis (Prete et al., 2021). Interestingly, EIF1AX mutations could co-occur with RAS mutations in ATC with a positive feedback relationship between RAS and EIF1AX proteins, which reinforces c-MYC gene ex-pression (Krishnamoorthy et al., 2019; Landa et al., 2016). Molecular alteration of the Wnt signaling pathway could be also observed notably with β-catenin gene (CTNNB1), AXIN1 and APC mutations (Prete et al., 2021). Although increased levels of cytoplasmic β-catenin are observed in most thyroid cancer cells, mutations of β-catenin that lead to nuclear localization of the protein are limited to PDTC and ATC suggesting a role in tumor progression (Garcia-Rostan et al., 2001).
Comment 2: Some typos are present:
Line 124: PDL-L1 (corrected form PD-L1)
Line 138: TAM (corrected form TAMs)
Line 172: PECT/TC (corrected form PET/TC)
We apologize for the typos. The modifications have been done.

Reviewer 2 Report
This manuscript reviewed the clinicopathologic features and current management of anaplastic thyroid carcinoma. The draft could be largely improved by scientific editing. Please also refer to the comments below.
1. Please use scientific description instead of "dramatic prognosis", "brutal onset", "local and distant evolution", "explosive transformation", etc.
2. Line 74, missing ")"
3. Line 74-76. The reference is old and applies the polygonal anti-PAX8 antibody, which most of the labs do not use anymore today. PAX8 stained only 54.4% of ATC using the most common MRQ50 monoclonal antibody in a large multi-institutional study consisting 182 cases (PMID: 31732814).
4. Line 80-82. Additional TP53 and/or TERT promoter mutations should be higher, in up to 96.3% of the cases.
5. Figure 4 is misleading in the sense that not all the BRAF/NTRK/RET/ALK+ ATCs have PD-L1 expression > 5% and are stage IVb or IVc. Please modify this figure. Specifically, the stage and PD-L1 parts may be better separated.
Author Response
- ANSWERS TO THE COMMENTS BY REVIEWER 2
Reviewer #2: This manuscript reviewed the clinicopathologic features and current management of anaplastic thyroid carcinoma. The draft could be largely improved by scientific editing. Please also refer to the comments below.
Comment 1: Please use scientific description instead of "dramatic prognosis", "brutal onset", "local and distant evolution", "explosive transformation", etc.
We have changed these expressions:
- Line 41 “poor prognosis” instead of “dramatic prognosis”
- Line 42 “rapid onset” instead of “brutal onset”
- Line 43 “with local and distant metastasis progression” instead of "local and distant evolution"
- Line 78: we have deleted the “explosive” expression
Comment 2: Line 74, missing ")"
We apologize for the typo. The modifications has been done.
Comment 3: Line 74-76. The reference is old and applies the polygonal anti-PAX8 antibody, which most of the labs do not use anymore today. PAX8 stained only 54.4% of ATC using the most common MRQ50 monoclonal antibody in a large multi-institutional study consisting 182 cases (PMID: 31732814).
Thank you for the comment. We have modified the text and added this reference:
“This study confirmed that thyroglobulin and TTF1 immunohistochemistry is almost always negative (96 and 70% of the cases, respectively while cytokeratin AE1/AE3 are present in 67% of the cases and PAX 8 in up to 70% (with anti-PAX8 antibody 10336-1-AP). A recent immunohistochemical study, us-ing withthe most commonly used monoclonal anti-PAX8 antibody (MRQ-50), showed lower PAX8 ex-pression in 54.4% of the ATC cases (Lai et al., 2020). This underlines the importance to perform Therefore, performing PAX8 immunohistochemistry in all samples of thyroid undifferentiated tumor sus-picious for ATC and in particular in squamous subtypes permit to support the differential diagnosis with squamous cell carcinoma of the head and neck which is always negative for PAX8 (Bishop et al., 2011).
Comment 4. Line 80-82. Additional TP53 and/or TERT promoter mutations should be higher, in up to 96.3% of the cases.
Thank you for this comment. In the study of Pozdeyev et al (Clin Cancer Res, 2018) the presence of TP53 and/or TERT promoter mutations was between 81.1% (159/196) and 88.3% (173/196) (14 patients tested for TP53 but not for TERT, table S6). Landa et al (J Clin Invest, 2016) have found an additional TP53 and/or TERT promoter mutations ratio of 97% (32/33 ATC patients). We propose to modified the text as follow:
“From a molecular point of view the association of additional TP53 and/or TERT promoter mutations is found in up to 80% of the ATC cases …”
Comment 5. Figure 4 is misleading in the sense that not all the BRAF/NTRK/RET/ALK+ ATCs have PD-L1 expression > 5% and are stage IVb or IVc. Please modify this figure. Specifically, the stage and PD-L1 parts may be better separated.
The figure 4 has been modified as suggested for a better under